# Perceptions of Body Condition, Diet and Exercise by Sports Dog Owners and Pet Dog Owners

**DOI:** 10.3390/ani11061752

**Published:** 2021-06-11

**Authors:** Heidi A. Kluess, Rebecca L. Jones, Tekla Lee-Fowler

**Affiliations:** 1School of Kinesiology, Auburn University, 301 Wire Rd, Auburn, AL 36849, USA; rlj0011@auburn.edu; 2Department of Clinical Sciences, College of Veterinary Medicine, Auburn University, 1200 Wire Road, Auburn, AL 36849, USA; tml0005@auburn.edu

**Keywords:** sports, pet, body condition, overweight, feeding, exercise

## Abstract

**Simple Summary:**

Dog obesity is a serious problem around the world. We investigated factors that contribute to dog obesity in pet dogs versus dogs that participate in sports with their owners. We found that pet dogs were more obese than pet dogs that participate in canine sports. This relationship had several contributing factors. One such factor is the ability of the pet owner to correctly identify the body condition of the dog. Inability to identify a dog as overweight/obese is a critical barrier to intervening in their dog’s health. Pet dogs were also fed more per feeding and given larger treats than the sports dogs. This increase in calories per day over time will result in weight gain. Lastly, many of our participants walked regularly with their dogs, but pet dog owners were more likely to report dog activities that did not directly involve the owner. This likely results in less exercise than that predicted by the owners. Overall, pet and sports dog owners put a high value on their dog’s health and well-being, but better education with regard to body condition, feeding and exercise is critical to improve the pet obesity problem.

**Abstract:**

The purpose of this study was to investigate the variables that contribute to obesity in pet dogs. The working hypothesis was that sports dog owners will better estimate their dog’s body condition and report stronger belief and control over their dogs’ feeding and exercise compared to traditional pet owners. We collected data on 171 pet owners (101 participated in canine sports) for this study. Each owner completed the Dog Owner Attitude Questionnaire. Each dog was measured for percent fat and Purina body condition scale. For the pet dogs, the median Purina body condition score was 6 (too heavy), but for the sports dogs it was 5 (ideal) (*p* < 0.05 different from pet dogs). The average percent fat for the pet dogs was 19.1 ± 8.6%, and for the sports dogs it was 13.8 ± 5.3% (*p* < 0.05 different from pet dogs). Among pet owners, 52% were able to correctly estimate their dog’s body condition. Sports dog owners were 57% correct. Pet dog owners fed approximately 60% more per day compared to sports dog owners. Pet and sports dog owners exercised their dog via walking, but sports dog owners reported more activity with the dog, while pet dog owners reported more activity than the dog did by themselves. Overall, pet and sports dog owners put a high value on their dog’s health and well-being, but better education with regard to body condition, feeding and exercise is critical to improve the pet obesity problem.

## 1. Introduction

Thirty-seven percent of US households own a dog, and recent statistics from the US, UK, Europe, and Australia suggest that around 50% of those pet dogs are obese [1,2,3,4,5]. All studies to date have focused on pet owners that are, for the most part, unsuccessful at maintaining their dog’s body condition at a healthy level [4,6,7,8,9,10,11,12]. While it is certainly important to understand why these pet owners act the way they do, sometimes just looking at pet owners that have successfully maintained their dog’s body condition provides an alternative viewpoint that may inform about the best course of action for interventions to reduce pet obesity.

Based on anecdotal observation of the primary investigators over many years working with pet owners and sporting dogs, the owners that participate in dog sports rarely have overweight or obese dogs. However, this is only anecdotal evidence, and there is no literature to date regarding the strategies and motivations these pet owners use to maintain their dogs’ body condition and where they get their feedback and information about the correct body condition, feeding and exercise for their dog. This unique group of pet owners is growing rapidly, with the American Kennel Club reporting approximately 3 million entries in 22,000 performance events in 2018.

Sports dog owners are generally leisure enthusiasts since these sports rarely have monetary awards associated with winning, and participation is quite costly, including entry fees, travel, training and equipment. People that participate in dog sports have a high level of autonomy and strong views about the behaviors that they engage [13,14], and these views may drive sports dog owners to be more knowledgeable about dog feeding and exercise. Gillespie et al. [15] described dog sports as a “culture of commitment,” which includes money and time. This ‘culture of commitment’ also includes a strong connection with other dog sports participants [15] that may include sharing information about feeding, body condition of dogs, and exercise for the dog that is sports and non-sports specific.

Furthermore, to properly intervene and provide a dog with a healthy environment to maintain weight, it is important to understand the mechanisms of change. The Theory of Reasoned Action has been used to understand how attitudes, subjective norms and perceived behavioral control relate to the feeding and exercise behaviors of dog owners [10,16]. The Theory of Planned Behavior is used to understand primarily human behaviors. The working hypothesis of this study is that sports dog owners will better estimate their dog’s body condition and report stronger belief and control over their dogs’ feeding and exercise compared to traditional pet owners. We also expect that sports dog owners will be more influenced in their feeding and exercise behavior by other dog owners, while traditional pet owners will be more influenced by their veterinarian. We expect that the main correlate with body condition for all dogs will be amount fed per day and a smaller relationship with exercise in both groups.

## 2. Methods

### 2.1. Participants

Pet dogs and their owners were recruited using flyers from well-dog visits at the Auburn University Veterinary School and around the local area. We also recruited using flyers on social media. Canine sports dogs and their owners were recruited using flyers at dock diving, water trial, scentwork and agility events in Alabama, Georgia and Missouri. The researchers attended these events and collected data on site. The inclusion criterion for the dogs was that they must be between 18 months and 10 years old and in good general health as reported by the owners. In addition, sports dogs had to be currently competing in a trial and have at least one sport title. There were no inclusion/exclusion criteria for humans. Each person could only report on one dog.

### 2.2. Body Condition/Composition

We measured the dog’s body condition using the Purina body condition scale and anthropometric measurements using a tape measure. For the Purina 9-point body condition scale, the dog was assessed by a person with experience in assessing body condition using that scale. Body fatness (% fat) was calculated from anthropometric measurements using the formulas developed by Mawby and Bartges [17]:Males: % fat = −1.4 × (hock to stifle length_cm_) + 0.77 × (pelvic circumference) + 4(1)
Females: % fat = −1.7 × (HS) + 0.93 × (pelvic circumference) + 5. (Measurements were obtained with a flexible tape measure.)(2)

### 2.3. Modified Dog Owner Attitude Questionnaire

The owner was asked to answer a questionnaire based on the Dog Owner Attitude Questionnaire developed by Rohlf et al. [10], which includes self-reported feeding behaviors, self-reported exercise behaviors, the owner’s opinion of the dog’s body condition, and exercise and feeding intentions. The survey was designed based on Theory of Planned Behavior and assessed intentions, subjective norms, perceived behavior control and behavior. We added a question, “What do you use to evaluate your dog’s body condition? (check all that apply) [Veterinarian; Friends; Dog trainer; Articles in magazines or on the internet; Breeder].” The purpose of this question is that we suspect that the source of information on body condition is different for sports dog owners compared to pet owners. We also added descriptive questions for sports dog owners to capture how many sports the dog is involved in.

### 2.4. Data Analysis

All descriptive data were summarized using mean ± standard deviation or the median and range (depending on the type of data). Internal consistency of the survey was assessed with Cronbach’s alpha > 0.67. Questionnaire questions were categorized as beliefs, intentions, perceived control or norms and composite scores created. A *t*-test was run to determine differences and a Cohen’s *D* effect size test was run to determine relative importance. For some variables, we ran a Pearson product–moment correlation using GraphPad Prism 7.0. The projected sample size with an alpha level of 0.05 and a power of 0.80 is 70 per group with a total sample size of 140.

## 3. Results

### 3.1. Human Participants

Overall, we received consent from 175 pet owners (70 pet owners and 105 sports dog owners). We had two participants that declined to get their dogs measured and two participants that chose not to participate in the study after signing the informed consent. Pet owners were younger (37 ± 16 years) than sports dog owners (54 ± 16; *p* < 0.05). Pet owners and sports dog owners were predominantly women (73% women in the pet dog group; 86% women in the sports dog group). Other information about the owners is included in Appendix A.

### 3.2. Dog Participants

The dog ages for both groups were similar (pet dog: 5 ± 3; sports dog: 5 ± 2 years). The pet dogs were 57% female and the sports dogs were 52% female. A larger number of pet dogs were desexed (spayed/neutered: 87%) compared to the sports dogs (spayed/neutered: 64%). Pet dogs were 53% mixed breeds, but sports dogs were only 11% mixed breeds. More information about dog breeds is included in Appendix A. The owners were asked about the size of their dogs. The median response was “medium sized.” The distribution of dogs per size category is shown in Figure 1.

For the pet dogs, the median Purina body condition score was 6 (range: 4–9), but for the sports dogs it was 5 (range: 3–8) (*p* < 0.05 different from pet dogs). Using the Purina body condition scale, 78% of pet dogs were classified as “too heavy” (score of 6–9) compared to 45% for the sports dogs. See Figure 2 for a distribution of Purina body condition scores by group.

The average percent fat for pet dogs was 19.1 ± 8.6%, and for the sports dogs it was 13.8 ± 5.3% (*p* < 0.05 different from pet dogs). The percent fat for all dogs spayed/neutered was 16.1 ± 7.0%, and it was 15.0 ± 7.7% for all intact dogs.

We asked all dog owners to rate their dog’s body condition (emaciated, underweight, optimal, overweight or obese) and compared what they reported to the Purina scale body condition score measured by the researchers. For pet owners, 52% were able to correctly estimate their dog’s body condition. Sports dog owners were slightly better at estimating their dog’s body condition with 57% correct estimates.

We asked participants where they got the information about their dog’s body condition. Pet owners and sports dog owners used their veterinarian as a source of information (pet: 96%; sports: 65%). Most pet owners (86%) reported only one source of information about the dog’s body condition, but only 51% of the sports dog owners reported using only one source of information. Sports dog owners reported use of friends (23%), dog trainers (34%), articles in magazines (14%) and the breeder (22%). When asked if it was important that their dog was in the correct body condition, both groups mostly reported “extremely important” (123/168 owners). The correlation between the Purina body condition score and the percent fat calculation was weak/moderate (*r* = 0.58, *p* < 0.0001).

### 3.3. Feeding

In the pet dog group, 87% of the owners reported feeding dry kibble, but only 60% of sports dog owners reported feeding dry kibble. The most commonly reported brand for both groups was Purina (pet: 25%; sports: 23%). There were a variety of brands written in. See the Appendix A for more details. The most common brand in the pet dog group was Royal Canin (7/67 dogs), but the most common in the sports dog group was Fromm (7/100 dogs). Out of the 44 brands written in (both groups combined), only 12 of the brands were reported by both groups. The sports group had a large percentage that wrote in feeding a raw diet (23%), but the raw diet was not reported at all in the pet dog group. A combination diet (wet/dry or raw/dry) was more commonly reported in the sports dog group (sports: 11%; pet: 3%).

The median amount fed was 2–2.5 cups per day for the pet dogs and 1.5–2 cups per day for the sports dogs. The times per day to feed was the same for both groups, at two times per day. The size of treats was 14.4 ± 17.6 cm^2^ for pet dogs, but 5.3 ± 12.7 cm^2^ for sports dogs (*p* < 0.05 different from pet dogs). The number of treats per day was the same for both groups (3–4 treats a day). We ran a Pearson product–moment correlation, assessing the relationship between the size of dog (owner-reported) and the reported amount of food fed per day. For pet dog owners, the relationship between owner perceived size of dog and feeding was strong (*R* = 0.71, *p* < 0.05), but less strong for the sports dog owners (*R* = 0.38, *p* < 0.05).

### 3.4. Exercise

The average times per week people exercised their dogs was not different among the groups (pet: 7.5 ± 6.0; sport: 7.0 ± 3.5 times per week). The main type of exercise reported for pet dog owners was walking on lead (64%), while the sports dog owners reported playing fetch or other games (55%). See Figure 3 for activities by group.

In the pet dog group, 43% reported multiple exercise activities, while 50% of the sports dog group reported multiple exercise activities. This question had a write in portion and 16 of the pet dog people and 74 of the sports dog people answered. Pet dog owners were more likely (88%) to report exercise that did not involve the owner (free running in yard, playing with other dogs) compared to sports dog owners (12%). The median length of time exercised was 15–30 min per exercise session for both groups. When asked about how much inactivity contributed to a dog becoming overweight/obese, most owners reported “very much” (117/168 owners).

### 3.5. Questionnaire

For the questionnaire, we only included composite scores that had a Cronbach’s alpha of at least 0.67. The questions associated with each category are included in Appendix A.

### 3.6. Feeding

The summary of the feeding portion of the questionnaire is shown in Table 1. See Appendix A for the questions included in each section.

In general, all owners felt that the barriers to correct feeding were low with the pet owners somewhat less confident than sports dog owners (*p* < 0.0001). Beliefs about feeding control were very high in both groups with the sports dog owners having more confidence than pet owners (*p* = 0.037). The reliance on other dog owners for feeding norms was low in both groups. However, pet owners reported a strong reliance on the veterinarian for feeding norms, but the sports dog owners were more neutral about the veterinarian as a source of feeding information (*p* = 0.0001). For all owners, their dogs being healthy was “extremely important.” The idea that too much food contributed to a dog becoming overweight/obese was also “very much” for owners (129/168 owners).

### 3.7. Exercise

The summary of the exercise portion of the questionnaire is shown in Table 2. The questions associated with each category are given in the Appendix A.

External beliefs about exercise were generally low among sports dog and pet owners, with sports dog owners having the lowest scores (*p* = 0.001). Negative exercise beliefs (You do not exercise your dog enough) were low in both groups, with the sports dog group having the lowest response (*p* = 0.002). The belief that exercise has value for the dog was high in both groups, but was slightly higher in the sports dog group (*p* = 0.017). The feeling of control over the amount and type of exercise their dogs received and the intention to exercise their dogs in the future were also high in both groups, but the highest in the sports dog group (*p* < 0.007). The participants generally disagreed that they lacked knowledge about the correct exercise for their dogs, but the pet dog group was less confident (*p* = 0.0001). Both groups felt that the views of other dog owners were not important to them with regard to exercising their dogs. The pet dog group felt that the veterinarian’s recommendations were important to them. The sports dog participants were more neutral about the role of the veterinarian (*p* < 0.0001).

## 4. Discussion

This is the first study to investigate the perspectives of sports dog owners in the maintenance of body condition in dogs. We found that more pet dogs that did not participate in sports were “too heavy” compared to pet dogs that did participate in sports. However, only 50–60% of the owners were able to correctly identify their dog’s body condition. This strongly suggests that pet and sports dog owners need better information regarding identifying correct body condition for dogs. The higher incidence of obesity in pet dogs was likely explained by the larger feeding volume and the greater number of treats. Exercise may also contribute because pet owners reported less variety of exercise activities and greater reliance on exercise that did not involve the humans. Sports dog owners had a wide variety of information sources for feeding and exercise. However, pet owners put a heavy reliance on the advice of the veterinarian, which underscores the importance of feeding and exercise counseling by the veterinarian to mitigate obesity in pet dogs.

### 4.1. Dogs

One of the most relevant findings was that sports dogs are leaner than pet dogs. Using the Purina body condition scale, 78% of pet dogs were classified as “too heavy” (score of 6–9) compared to 45% of the sports dogs. Around the developed world, it is estimated that 40–60% of pet dogs are obese [2,7,18]. Eastland-Jones et al. [19] found that 91% of pet dogs were classified by the investigator as “overweight.” Rohlf et al. [10] found that 46% of their cohort was overweight, but only 10% were classified as obese. 

### 4.2. Spay/Neuter Status

There are a number of factors that likely contributed to leaner dogs in the sports group compared to the pet group, including the spay/neuter status and owner’s understanding of correct body condition, feeding and exercise. One interesting finding was that a larger number of pet dogs were neutered or spayed compared to the sports dogs. This may be due to the recent increase in information about the benefits of maintaining your dog intact for muscle growth and prevention of injuries [20,21,22,23]. There is research suggesting that neutering/spaying a dog is a correlate for obesity [2,4]. However, McGreevy et al. [1] showed no relationship between spay/neuter status and obesity, but 75% of their sample was spayed or neutered. Lund et al. [2] reported a higher prevalence of overweight in spayed and neutered dogs. Courcier et al. [4] reported that female spayed dogs had a higher risk of obesity compared to male neutered dogs. Interestingly, Bjornvad et al. [5] reported a higher incidence of obesity in neutered male dogs, but not in female dogs. A review by Linder and Mueller [24] reported neutering/spay as a risk factor worldwide for pet obesity. In the current study, when combining the groups and looking at percent fat in spayed/neutered vs. intact, we saw no differences, suggesting that other factors may be more important than spay/neuter status in the current study. We also did not see differences in percentage fat when sex and spay/neuter status were combined.

### 4.3. Owner’s Knowledge of Dog’s Body Condition

We asked all dog owners to rate their dog’s body condition (emaciated, underweight, optimal, overweight or obese) and compared what they reported to the Purina scale body condition score measured by researchers. For pet owners, 52% were able to correctly estimate their dog’s body condition. Sports dog owners were slightly better at estimating their dog’s body condition with 57% correct estimates. These correct body composition estimates are similar to those reported by Courcier et al. [4] but lower than those of Eastland-Jones et al. [19]. White et al. [12] also assessed owner versus veterinarian assessment of body condition. They found that the owner and the veterinarian generally agreed when the dog was underweight or normal weight, but the agreement was only 53% when the dog was overweight. This finding is supported by Rohlf et al. [10], who found a similar result. Webb et al. [25] reported that 33% of their sample reported their dog’s body condition as normal when an expert rated them as overweight. This research suggests that understanding of correct dog body condition is relatively poor in all pet-owning groups.

Correct estimation of the dog’s body condition is critical for making improvements in the dog’s health; therefore, we asked participants where they got their information about their dog’s body condition. Pet owners and sports dog owners used their veterinarian as a source of information. Most pet owners reported only one source of information about the dog’s body condition, but only 51% of the sports dog owners reported using only one source of information. The sports dog owners reported use of friends, dog trainers, articles in magazines and the breeder. It is clear that both groups place a high value on their dog’s physical health, but it seems that relevant information about their dog’s body condition is not completely addressed by the veterinarian. Rolph et al. [26] investigated whether veterinarians recorded if dogs were overweight. In almost 50,000 consultations with owners of overweight dogs, only 1.4% were recorded as overweight by the veterinarian. White et al. [27] found that 80% of owners reported that they had discussed the issue of their dog’s weight with their veterinarian and 69% knew their dog’s weight. Anecdotally, we noticed that many dog owners (particularly sports dog owners) were aware of the dog’s weight on a scale, suggesting that many veterinarians are recording and sharing this information; however, fewer participants were aware that body condition could change even though the dog’s weight remained the same. Using a pictorial and written scale, such as the Purina body condition scale, may be a better teaching tool to help people to monitor their dog’s body condition.

### 4.4. Feeding

Pet dog owners fed their dogs approximately ½ cup more food per day compared to the sports dog owners. The times per day to feed was the same for both groups, at two times per day. Kienzle et al. [9] reported higher numbers of meals and snacks in obese dogs, compared to normal-weight dogs. Bland et al. [7] found that owners of obese dogs fed more per feeding than owners of non-obese dogs. In the current study, the size of treats was approximately 60% larger in pet dogs versus sports dogs. Overall, the combination of higher amounts of food per feeding and larger-sized treats suggests that pet dog owners are feeding their dogs more calories per day, which, in part, may contribute to more overweight dogs in the pet group.

In general, all owners felt that the barriers to correct feeding were low, with the pet owners somewhat less confident than the sports dog owners. Beliefs about feeding control were very high in both groups, with the sports dog owners having more confidence than the pet owners. The pet owners reported a strong reliance on the veterinarian for feeding norms, but the sports dog owners were more neutral about the veterinarian as a source of information about feeding. For all owners, their dog being healthy was “extremely important. The idea that too much food contributed to a dog becoming overweight/obese was also “very much” for owners. Owners have a strong desire to feed their dogs correctly, but pet owners need more guidance on how much to feed and the amount of treats given.

### 4.5. Exercise

Participants reported exercising their dogs 7 days per week for 15–30 min per session. The main type of exercise reported for pet dog owners was walking on lead, while the sports dog owners reported playing fetch or other games. In the pet dog group, 43% reported multiple exercise activities, while 50% of the sports dog group reported multiple exercise activities. Pet dog owners were more likely to report exercise that did not involve the owner (free running in the yard, playing with other dogs) compared to sports dog owners. When asked about how much inactivity contributed to a dog becoming overweight/obese most owners reported ‘very much.’ Therefore, owners have a high value for exercise as a health benefit to their dog, but may not have a good gauge for how much exercise the dog is actually getting. Bland et al. [7] found that owners with overweight dogs were more likely to be confined to a yard for exercise, rather than walked. Rohlf et al. [10] also found walking the dog as the most popular exercise activity. Dog walking may also help reduce canine obesity. Warren et al. [28] used accelerometers on dogs and showed that a higher number of steps walked was related to a lower body condition score (leaner).

External barriers to exercise were generally low among sports dog and pet owners, with sports dog owners having the lowest scores. Negative exercise beliefs (you do not exercise your dog enough) were low in both groups, with the sports dog group having the lowest response. As with feeding, both groups felt that the views of other dog owners were not important to them with regard to exercising their dogs, but the pet dog group felt that the veterinarian’s recommendations were important to them. The sports dog participants were more neutral about the role of the veterinarian. These results underscore the importance of the veterinarian in providing good information about exercise for pet dogs.

### 4.6. Limitations

One limitation of this study is the use of owner-reported physical activity. It is well understood that people tend to overestimate the amount of physical activity in which they participate [29]. However, questionnaires are widely used for estimating physical activity in dog owners [10,30,31,32,33].

Another limitation is that we used convenience sampling for recruitment. This may have resulted in a larger portion of people volunteering to participate that had lean dogs. However, our data were consistent with other studies and our research suggests that many people underestimate their dog’s body condition. The ability of a person to access their veterinarian to receive help in correctly estimating their dog’s body condition is limited by income [34]. In the current study, our participants had medium to high socioeconomic status (see Appendix A). However, future consideration should be given to low-socioeconomic pet owners and how they maintain their dog’s body condition.

This is the first study investigating the body condition of dogs that participate in sports. Surprisingly, we found that 45% of the sports dogs were classified as “too heavy.” This may have a significant impact on lifelong sports involvement, particularly in high-impact sports. One of the most significant factors in sports dog obesity was the inability of the owner to identify correct body condition in their dog. Future work needs to focus on the best method to educate sports dog owners about body condition. A more complete investigation of the value of sports participation in the human side of the dog sports teams is also warranted.

For the current study, we concluded that pet dogs were more obese than pet dogs that participate in canine sports. This relationship had several contributing factors. One such factor is the ability of the pet owner to correctly identify the body condition of the dog. Inability to identify the dog as overweight/obese is a critical barrier to intervening in their dog’s health. Pet dogs were also fed more per feeding and given larger treats than sports dogs. Lastly, many of our participants walked regularly with their dogs, but pet dog owners put a higher value on dog activities that did not directly involve the owner. This likely resulted in less exercise than that predicted by the owners. Overall, pet and sports dog owners put a high value on their dog’s health and well-being, but better education with regard to body condition, feeding and exercise is critical to improve the pet obesity problem.

## Figures and Tables

**Figure 1 animals-11-01752-f001:**
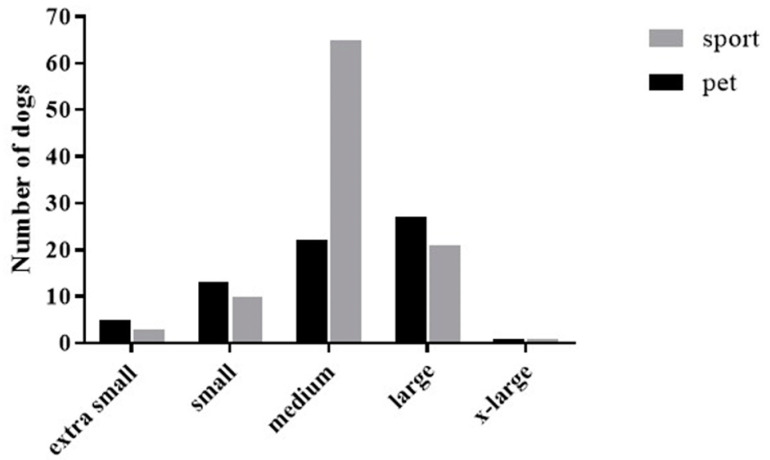
Size distribution of dogs according to owner report.

**Figure 2 animals-11-01752-f002:**
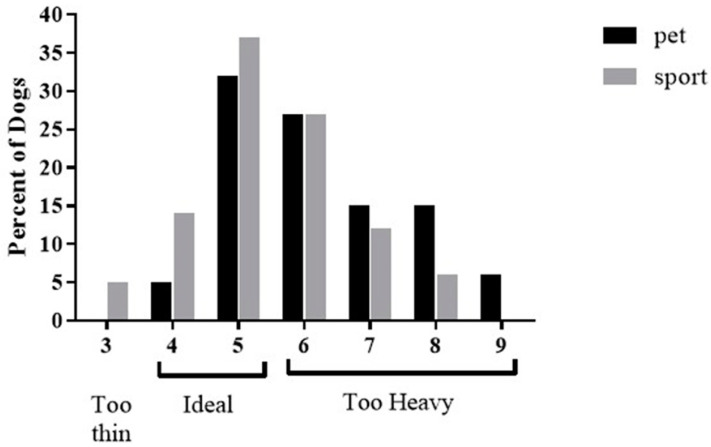
Percentage of dogs for different Purina Body Condition scores in the pet dog group (dark bars) and the sport dog group (light gray bars).

**Figure 3 animals-11-01752-f003:**
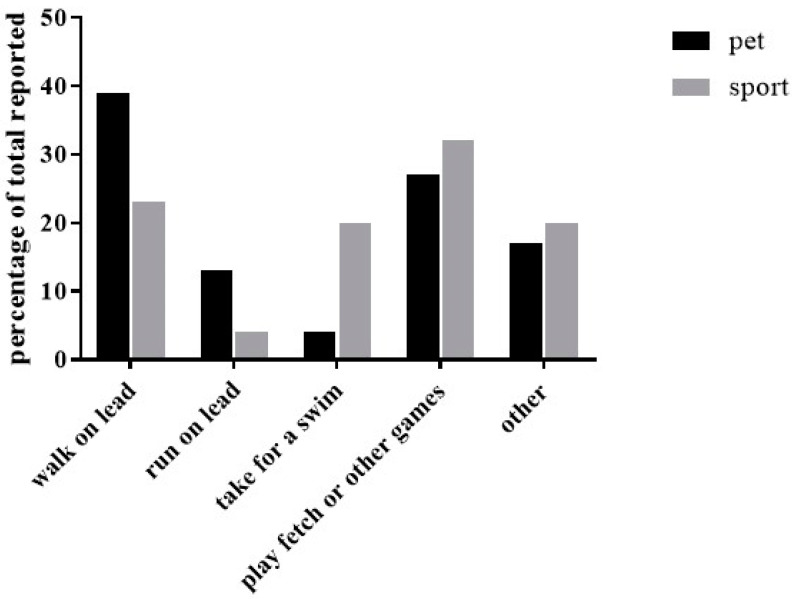
Percentage of owners that reported different types of exercise activities by group.

**Table 1 animals-11-01752-t001:** Beliefs and intentions about feeding.

Categories from the Questionnaire	Average Score of Sports Dog Group *n* = 101	Average Score of Pet Dog Group *n* = 70	*p*	Effect Size Cohen’s D	Cronbach’s Alpha
Feeding barriers, external owner	1.7 ± 0.7	2.2 ± 1.0	0.0001	−0.57	0.71
Feeding control	6.7 ± 0.8	6.4 ± 0.9	0.037	0.33	0.71
Feeding norms comply with others	2.9 ± 1.3	2.7 ± 1.2	NS	0.16	0.79
Feeding norms comply with vet	4.1 ± 1.4	6.0 ± 1.0	0.0001	−1.55	0.86
Feeding norms, other	2.0 ± 0.9	2.3 ± 0.9	0.024	−0.35	0.76

Mean ± Standard Deviation.

**Table 2 animals-11-01752-t002:** Beliefs and intentions about exercise.

Categories from the Questionnaire	Average Score of Sports Dog Group *n* = 101	Average Score of Pet Dog Group *n* = 70	*p*	Effect Size Cohen’s D	Cronbach’s Alpha
Exercise beliefs external	1.6 ± 0.8	2.1 ± 1.1	0.001	−0.51	0.85
Exercise beliefs owner	1.8 ± 0.8	2.2 ± 0.8	0.002	−0.46	0.78
Exercise beliefs value	6.5 ± 0.6	6.3 ± 0.7	0.017	0.38	0.72
Exercise control	6.6 ± 0.9	6.2 ± 1.0	0.007	0.43	0.95
Exercise intentions	6.6 ± 0.7	6.2 ± 0.8	0.0001	0.56	0.89
Exercise lack of knowledge	2.0 ± 1.0	3.2 ± 1.5	0.0001	−0.92	0.85
Exercise norms comply with others	3.4 ± 1.4	3.2 ± 1.5	NS	0.14	0.85
Exercise norms comply with vet	4.6 ± 1.2	6.1 ± 0.8	0.0001	−1.4	0.89
Exercise norms other	2.0 ± 1.0	2.6 ± 1.0	0.00001	−0.66	0.89

Mean ± Standard Deviation.

## Data Availability

The data is available here https://hak0006.wixsite.com/vascularphyslab/data (accessed on 11 June 2021).

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
