# Peer review of "Perceptions of Body Condition, Diet and Exercise by Sports Dog Owners and Pet Dog Owners"

_animals, 2021, doi:10.3390/ani11061752_

Round 1

Reviewer 1 Report

The presentation is generally satisfactory but two things need to be resolved

In my first look at this paper I queried how the Mawby and Bartges equation to predict body fat could take into account phenotypic differences in body:leg length (e.g. dachshund v. Saluki).  This question has not been addressed in this draft. However now  there are no results for body fat estimates based on this scale, although the equation remains in the text. I assume you have discarded it so take it out.

In discussion you state that there were no significant effects of neutering in your study. However, you must present the actual values in your results. I suggest a table of the form

                                 Male                Female

                       Entire   Neutered   Entire    Neutered

       Sport

       Pet      

Author Response

I added the spay/neuter status percent fat to the results in lines 141-2

Reviewer 2 Report

This is interesting work, especially the impact of vets on pet owners vs. sport dog owners

Title: too broad, revise to better reflect the work performed e.g. Perceptions of body condition, diet and exercise by sport dog owners vs. pet dog owners.

Line 60: edit ‘ to “ in front of culture of commitment”

Line 64-66: sentence is hard to follow on first read; revise this sentence to clarify or be more concise

Line 66: should theory of reasoned action be capitalized?

Line 71: defining/explaining what is meant by stronger belief and stronger control would be helpful

Line 74-75: The last sentence/hypothesis is hard to follow, are you suggesting that amount fed will have a greater correlation with BCS, and exercise will have a weaker correlation with BCS?

Methods: there is no mention of owner BMI in this section yet you have data in the supplementary data, add BMI methods to main text, also need to mention owner BMI in methods then you can reference your supplementary data

Methods & Figure 1: dog body weight should be included in this study, as objective data of dog size; relying on owner perception of dog size alone is weak, especially since you are making assumptions about what the food volume data means

Methods: did you assess muscle condition score, along with BCS? At line 289-290, you state that body condition can change without a change in body weight, I assume you are referencing a shift in fat and muscle mass, thus muscle condition would be relevant to assess in this study, if nothing else to subjectively determine that the sport and pet dogs had normal muscle condition

Results: move client income data from the supplementary data into the main text, to join client gender data

Results: reference your supplementary data in the main text (e.g. dog breed)

Line 142: revise the period in front of “dogs:25.6”

Line 167-169: food/treat volume/size data is not helpful without dog body weight data and the energy density of the foods/treats; while the latter would require an extensive review of the diet history that seems outside the scope of this study, including body weight data is an important compliment to the owner perception of dog size data … also, did you categorize the type of treats or just treat size? Size alone isn’t super helpful without knowing the type of treat

Line 173: add “owner perceived” in front of size, this needs to be complimented with dog body weight data

Tables 1 & 2: you are using codes yet not providing readers enough data to understand them, these need to be defined or at least include examples from the supplementary data in the main text; I’m not sure the tables are helpful as the data is also provided in text, which is redundant; what are the values of the last two columns?

Line 209: provide examples of external barriers to exercise

Lines 218-221: consider revising to read “both groups felt that the views of other dog owners, regarding exercise, were not important. The pet dog group felt that the veterinarian’s recommendations, regarding exercise, was important. The sports dog participants were more neutral about the 220 role of the veterinarian in their dog’s exercise (p<0.0001).”

Revise section titles “Mean +/- standard deviation”, throughout the results & discussion, to reflect the content

Line 224: add “more” in front of pet dogs

Line 224-225: overweight as defined by the owner or the investigator?

Line 228-229: without body weight data to confirm dog size, this conclusion is a stretch  

Discussion: inclusion of results in this section is redundant with the results section, save word count by removing results data in this section and sticking to discussing what the results mean

Discussion: it is unclear why there is a discussion section in the supplementary information, if it is due to word count limits then you can make room in the discussion section by eliminating the redundancy of results in the discussion section, move the client BMI, dog BCS and dog feeding discussions to the main text

Line 238-239: delete sentence because overweight incidence was already covered in the discussion

Line 301: you did not measure calorie intake, thus this is a stretch; overweight dogs are overfed calories relative to their individual need, but that doesn't mean that they are fed more than non-overweight dogs as energy needs vary widely across individuals

Limitations: without body weight, you can’t confirm that owner-assessed dog size is a major limitation

Line 361: you cannot assume kcal intake when it wasn’t measured

References: move references in supplementary data into main text

Author Response

Title: too broad, revise to better reflect the work performed e.g. Perceptions of body condition, diet and exercise by sport dog owners vs. pet dog owners.

changed

Line 60: edit ‘ to “ in front of culture of commitment”

This doesn’t make sense

Line 64-66: sentence is hard to follow on first read; revise this sentence to clarify or be more concise

I just removed the sentence.

Line 66: should theory of reasoned action be capitalized?

done

Line 71: defining/explaining what is meant by stronger belief and stronger control would be helpful

It is from the likert scale. (SA, A etc)

Line 74-75: The last sentence/hypothesis is hard to follow, are you suggesting that amount fed will have a greater correlation with BCS, and exercise will have a weaker correlation with BCS?

Yes, that’s correct

Methods: there is no mention of owner BMI in this section yet you have data in the supplementary data, add BMI methods to main text, also need to mention owner BMI in methods then you can reference your supplementary data

I added “(calculated from owner reported height and weight)” to the BMI results in the supplementary data. I don’t think it is appropriate to include it in the main methods section.

Methods & Figure 1: dog body weight should be included in this study, as objective data of dog size; relying on owner perception of dog size alone is weak, especially since you are making assumptions about what the food volume data means

Dog body weight is not a typical method for determining whether a dog is medium sized or not. I would argue that body weight is an objective measure but, classifying a dog as medium sized by weight is equally problematic.

Methods: did you assess muscle condition score, along with BCS? At line 289-290, you state that body condition can change without a change in body weight, I assume you are referencing a shift in fat and muscle mass, thus muscle condition would be relevant to assess in this study, if nothing else to subjectively determine that the sport and pet dogs had normal muscle condition

This is a point of discussion regarding the use of the Purina BCS rather than just using body weight. It was not a comment on the results of this study.

Results: move client income data from the supplementary data into the main text, to join client gender data

I’m not clear about the value of this change. Line 124 I added…” Other information about the owners is included in the supplementary file.”

Results: reference your supplementary data in the main text (e.g. dog breed)

Line 130. I added “More information about dog breeds are included in the supplementary file.  “

Line 142: revise the period in front of “dogs:25.6”

done

Line 167-169: food/treat volume/size data is not helpful without dog body weight data and the energy density of the foods/treats; while the latter would require an extensive review of the diet history that seems outside the scope of this study, including body weight data is an important compliment to the owner perception of dog size data … also, did you categorize the type of treats or just treat size? Size alone isn’t super helpful without knowing the type of treat

Most owners did not provide just one type of treat, so this would really be impossible. I’m sorry that you think that size is not helpful and certainly some treats are more calorically dense than others, however, larger is likely more calories.

Line 173: add “owner perceived” in front of size, this needs to be complimented with dog body weight data

I changed the wording. I would be happy to include the weight data if you can show me weight categories by dog body size that you think are appropriate and have a reasonable reference associated.

Tables 1 & 2: you are using codes yet not providing readers enough data to understand them, these need to be defined or at least include examples from the supplementary data in the main text; I’m not sure the tables are helpful as the data is also provided in text, which is redundant; what are the values of the last two columns?

I added to line 192-3 and 208-9: “The questions associated with each category are in the supplementary files.” I really don’t know what to do with this at this point. In the last review you all wanted the questions (that helped you understand the categories) out of the main text. The categories and the questions are in the supplementary file. These are standard tables used in many questionnaire based studies. The values in the last 2 columns are the statistics for the sections. This is also standard practice.

Line 209: provide examples of external barriers to exercise

I changed it to “External beliefs about exercise”

Lines 218-221: consider revising to read “both groups felt that the views of other dog owners, regarding exercise, were not important. The pet dog group felt that the veterinarian’s recommendations, regarding exercise, was important. The sports dog participants were more neutral about the 220 role of the veterinarian in their dog’s exercise (p<0.0001).”

Changed

Revise section titles “Mean +/- standard deviation”, throughout the results & discussion, to reflect the content

Those sections were inserted by the editors. It was supposed to be part of the table. The paragraph below relates to the table results.

Line 224: add “more” in front of pet dogs

done

Line 224-225: overweight as defined by the owner or the investigator?

By the investigator.

Line 228-229: without body weight data to confirm dog size, this conclusion is a stretch  

“Overweight” may be misleading. I changed it to “too heavy” to reflect the wording on the Purina scale.

Discussion: inclusion of results in this section is redundant with the results section, save word count by removing results data in this section and sticking to discussing what the results mean

In the places where I used some data (percentages) I was using the numbers to compare to other studies results. This is actually important to the discussion.

Discussion: it is unclear why there is a discussion section in the supplementary information, if it is due to word count limits then you can make room in the discussion section by eliminating the redundancy of results in the discussion section, move the client BMI, dog BCS and dog feeding discussions to the main text

In the last review it was requested that we remove that data and move the discussion to the supplementary file.

Line 238-239: delete sentence because overweight incidence was already covered in the discussion

This line is in the discussion section. Not sure what the issue is.

Line 301: you did not measure calorie intake, thus this is a stretch; overweight dogs are overfed calories relative to their individual need, but that doesn't mean that they are fed more than non-overweight dogs as energy needs vary widely across individuals

This is a discussion comment and I did say “suggest that pet dog owners are feeding their dog more calories per day” not that it was measured.

Limitations: without body weight, you can’t confirm that owner-assessed dog size is a major limitation

I changed the term “overweight” to “too heavy”

Line 361: you cannot assume kcal intake when it wasn’t measured

I removed that statement.

References: move references in supplementary data into main text

The references in the supplement reflect the references used in the supplement. The main text references reflect the citations in the main text.

This manuscript is a resubmission of an earlier submission. The following is a list of the peer review reports and author responses from that submission.

Round 1

Reviewer 1 Report

This paper addresses an important issue. At present, however, it has several major omissions.

The description of the data analysis needs to be improved. What do the first columns of numbers in Tables 1 and 2 refer to? e.g. Feeding barriers 1.7 v.2.2, Feeding controls 6.7 v. 6.4? The tables themselves can be better presented, listing only the 5 and 8 main categories respectively, with the subcategories in the text, or preferably in an appendix. What is the significance of Cronbach's alpha >0.67?

The equations used to estimate BMI and  Body fat index (amazingly) give no consideration to conformation differences between breeds. A healthy, lean bulldog and a whippet will have a very different BMI . Likewise the BFI for a Basset Hound and a Greyhound.  Surely someone must have taken these phenotypic differences into account. At presented , I suggest they are meaningless. This paper would be much more valuable if it recorded and took into account these things. For example, the fact that sport breeds were less obese according to the BFI, but there was no significant difference in BMI could be attributed entirely to differences in the conformation of the lean body mass.

The results and discussion need to be rearranged. Most of the results appear in the discussion.

Reviewer 2 Report

I am sympathetic to the effort of the authors to look for factors that contribute to obesity in pet dogs. The idea of comparing two groups of owners where the dogs differ in their levels of obesity across the groups is also fine. However, great care is needed when trying to draw conclusions from such a cross-sectional design.

There are a number of serious problems in the paper as it is:

Firstly there are things described in the methods section which are not used further on. For example it is said in l. 115-116 that the authors "added the owners height and weight so we could calculate owner-reported body mass index". However the these findings are not reported and therefore cannot be used as a possible explanation of the differences found in dog obesity (numerous previous studies have found a correlation between owner BMI and  dog obesity).

Secondly, the only main difference found between the two groups concerns feeding, where it is claimed that sports dog owner feed their dogs less than other pet dog owners. However, given the vastly different feeding regimes between the two groups described in l 174-182 it is highly unclear that any firm conclusions can be drawn about differences in calories fed. Also there seems to be no discussion of possible differences in sizes of dogs in the two groups.

Thirdly obvious confounders are not discussed. For example in l 141-143 it is described that the sports dogs are spayed and neutered to lesser degree than the other pet dogs. Here it would be relevant to bring in at recent Danish study (Bjørnvad CR, Gloor S, Johansen SS, Sandøe P & Lund TB. (2019). Neutering increases the risk of obesity in male dogs but not in bitches - A cross-sectional study of dog- and owner- related risk factors for obesity in Danish companion dogs. Preventive Veterinary Medicine, 170, 104730) that documents that neutering is major risk factor for obesity in male dogs. Also it is noted that the age of the two owner groups differ largely; and it would be relevant to discuss whether this or other demographic difference (such as education and income) could explain the differences found.

Finally, the paper contain findings, which are even mentioned in the abstract, that appear irrelevant. For example the non-validated BMI measure for obesity dogs is used, but it seems to deliver results that are at odds with results based on validated body condition and % fat. Why bring this in at all?

My main concern with the paper is that it in my view fails to deliver sound scientific evidence that can explain why one group is more lean than the other. And I cannot see that the paper makes a significant contribution to the already vast literature on dog obesity.

Reviewer 3 Report

General comments:

Discussion is missing any identification of future research directions or any particularly substantive reference to how these findings should inform future practice with general dog owners vs. sport dog owners.

Line 17 - Per your results, pet dog owners were just more likely to report using activities that did not involve the owner to exercise their dog (as opposed to placing a "larger value" on these types of activities. Please edit this language to accurately reflect your results.

Line 23 - Check grammar/English language use in the phrase "report stronger belief and control over their dogs' feeding and exercise" 

Line 42 - All the references to support the rationale for this work are over ten years old. Please add more recent research on this topic, if available.

Line 46 - The use of the phrase "successful pet owners" is problematic without specifically defining how/why sport dog owners are viewed as more successful than non-sport dog owners. If you are going to use this phrase to justify your comparison of sport dog owners to general dog owners, please add a specific definition of "success." 

Line 55 - Since these data were collected in 2019, are these AKC statistics available for the same year?

Line 56 - This term "leisure enthusiasts" is not common terminology, to my knowledge, so please define further.

Line 58-60 - Higher level of autonomy and strong views compared to who? This appears to be your primary support for the conclusion that sport dog owners might be "more successful pet owners," but you have not defined what constitute "successful pet ownership."

Line 66 - instead of using "we" please define who specifically needs to understand these mechanisms of change - veterinarians, pet owners, others?

Line 68-71 - This sentence reads rather awkward. Please review and edit for English language, grammar, and conciseness.

Line 75 - Since the word "belief" is not using anywhere in your instrumentation, it is difficult to differentiate which items in your results support this statement. Are you actually referring to the "intention" items? If so, please use "intention" here and in Line 23 instead of "belief."

Line 92 - Please define how "good general health" was determined for this inclusion criteria.

Line 125 - The Cronbach's alpha for the instrumentation is on the low end of the acceptable range, yet this is not discussed at all in the limitations section. Please add this to the limitation section of the Discussion.

Line 154 - Please report the p-value of this difference.

Line 175 - Rather than "median" I think you mean "most frequently reported." 

Line 220 - Please report out how this compared to the sport dog owners' responses.

Line 234 - Based on the instrumentation, I think you mean "of other dog owners"?

Line 239 - Rather than a study of "the role of dog sports in the maintenance of body condition in dogs", I view this as a study on how the perspectives of sport dog participants contribute to the maintenance of body condition.

Line 279 - No where in this discussion is it mentioned that access to veterinary care is extremely limited, particularly for low income pet owners. If a general pet owner is reliant on veterinarians for information on healthy body condition and maintenance for their pets, this major barrier to ensuring health/welfare of all pets needs to be acknowledged. LaVallee published a systematic review on this topic in 2017 that could be cited.

Line 320 - I think the appropriate word here is "gauge" not "gage."

Line 329 - Again, I think you mean to say "other dog owners"